# Moderate levels of 5-fluorocytosine cause the emergence of high frequency resistance in cryptococci

Yun C. Chang [1✉], Ami Khanal Lamichhane[1], Hongyi Cai[2], Peter J. Walter [2], John E. Bennett[3] & Kyung J. Kwon-Chung[1]

The antifungal agent 5-fluorocytosine (5-FC) is used for the treatment of several mycoses, but is unsuitable for monotherapy due to the rapid development of resistance. Here, we show that cryptococci develop resistance to 5-FC at a high frequency when exposed to concentrations several fold above the minimal inhibitory concentration. The genomes of resistant clones contain alterations in genes relevant as well as irrelevant for 5-FC resistance, suggesting that 5-FC may be mutagenic at moderate concentrations. Mutations in *FCY2* (encoding a known permease for 5-FC uptake), *FCY1*, *FUR1*, *UXS1* (encoding an enzyme that converts UDP-glucuronic acid to UDP-xylose) and *URA6* contribute to 5-FC resistance. The *uxs1* mutants accumulate UDP-glucuronic acid, which appears to down-regulate expression of permease FCY2 and reduce cellular uptake of the drug. Additional mutations in genes known to be required for UDP-glucuronic acid synthesis (*UGD1*) or a transcriptional factor *NRG1* suppress UDP-glucuronic acid accumulation and 5-FC resistance in the *uxs1* mutants.

[1] Molecular Microbiology Section, Laboratory of Clinical Immunology and Microbiology, NIAID, NIH, Bethesda, MD, USA. [2] Clinical Mass Spectrometry Core, NIDDK, NIH, Bethesda, MD, USA. [3] Laboratory of Clinical Immunology and Microbiology, NIAID, NIH, Bethesda, MD, USA. ✉email: ychang@niaid.nih.gov

5-fluorocytosine (5-FC), a fluorinated pyrimidine, is one of the oldest antifungals of use in treatment of cryptococcosis, as well as in candidiasis and chromoblastomycosis. It was synthesized in 1957[1] and is the only commercially available compound that inhibits macromolecule synthesis in fungi for clinical use. 5-FC enters fungal cells through one or more permeases among which purine-cytosine permease, Fcy2, is the primary one known for uptake of the drug[2–6]. 5-FC itself is not toxic and is converted to its metabolically active form 5-fluorouracil inside the cells by cytosine deaminase, Fcy1. 5-fluorouracil can be further processed to 5-fluorouridine monophosphate or 5-fluorodeoxyuridine monophosphate and inhibit DNA replication, transcription, and protein synthesis[4,5,7,8].

Currently, 5-FC in combination with amphotericin B remains the gold standard for induction therapy in cryptococcosis followed by maintenance therapy with fluconazole[9]. Combination therapy is known for more rapid cerebrospinal fluid sterilization and, though controversial, lower rates of treatment failure and improved mortality when compared with alternate regimens[10,11]. 5-FC, however, is not suitable for monotherapy due to rapid development of resistance in fungal pathogens. For instance, in the largest case series, only 10 of 23 (43%) cryptococcosis patients responded to flucytosine monotherapy[12]. From the 13 failing patients, isolates were available for study from 12 patients, among which six isolates were highly resistant to 5-FC. All six isolates had markedly decreased uptake of flucytosine and cytosine, suggesting an alteration in permease activity and five of them were cross resistant to 5-fluorouracil suggesting an altered uracil phosphoribosyltransferase[13]. However, this study was carried out before molecular technology was available and no subsequent study has been conducted on the clinically acquired 5-FC resistance in *Cryptococcus spp*. Resistance to 5-FC in other clinically relevant fungal species is known to emerge as a consequence of mutations primarily in the *FCY2*, *FCY1*, and *FUR1* genes encoding the purine-cytosine permease, cytosine deaminase, and uracil phosphoribosyltransferase, respectively[6,14–17]. Many more genes were found to contribute to 5-FC resistance through chemogenomic analysis in *Saccharomyces cerevisiae*[18] but the relevance of these genes in cryptococcal 5-FC resistance remains unclear. Recently, the *UXS1* gene, known to encode an enzyme that converts UDP-glucuronic acid (UDP-GlcUA) to UDP-xylose for capsule biosynthesis, has been reported to play a role in cryptococcal resistance to 5-FC in vitro[19].

Here, we show that various strains of *Cryptococcus neoformans* and *C. gattii* species complex develop resistant clones at very high frequency at the drug concentrations 20-to 100-fold of the minimal inhibitory concentration (MIC). Furthermore, the resistance phenotype is unstable in many of these clones. An array of mutations including insertion, deletion, substitution, and variations in chromosome copy number are identified via genomic sequencing of the resistant clones. Interestingly, the 5-FC resistance phenotype in the *uxs1* deletion mutants is unstable and we identify two genes whose mutations can independently suppress the 5-FC resistance phenotype of the *uxs1* mutants. We demonstrate that mutations in *UXS1* and the presence of its suppressors affect the levels of intracellular UDP-GlcUA. Importantly, accumulation of UDP-GlcUA appears to modulate the expression levels of cytosine permease Fcy2 which controls the accumulation of 5-FC with an outcome of altered drug susceptibility.

## Results

**5-FC resistant colonies emerge at high frequency at concentrations several folds above the MIC.** An interesting phenomenon was observed when we examined the 5-FC susceptibility in the strains of *Cryptococcus* species complex. The 5-FC resistant colonies emerged with frequency near $1 \times 10^{-1}$ at concentrations close to 20-fold above the MIC of H99, a patient isolate of *C. neoformans* VNI strain[20], R265 a Vancouver outbreak strain of *C. gattii* VGIIa[21], and WM276, a VGI reference strain of *C. gattii* isolated from Australian environment[22] (Fig. 1 and Supplementary Fig. 1). In each case, the frequency of resistance decreased dramatically at higher concentrations of the drug. Since all cryptococci thus far tested is known to develop unstable heteroresistance to fluconazole at high frequency[23–25], we determined if similar phenomena occurred with 5-FC. We randomly selected 15 each of large- and small-sized resistant colonies grown on 5-FC agar plates in which the resistance frequency was greater than $1 \times 10^{-3}$. The randomly selected clones were cultured in the nonselective YPD medium and transferred daily to fresh media. The frequency of 5-FC resistant population in each strain was determined periodically during the passage period. Close to 50% of the clones derived from the large-sized 5-FC resistant colonies remained resistant to the drug at the end of transfers in all three strains indicating that resistance to 5-FC in those clones were relatively stable (Supplementary Fig. 2, left panels). Considerably fewer number of clones derived from small-sized resistant colonies remained resistant at the end of passage. Furthermore, many of the unstable clones rapidly lost their resistance in less than seven passages regardless of their original colony size on the 5-FC media. These observations indicated that the stability of resistance

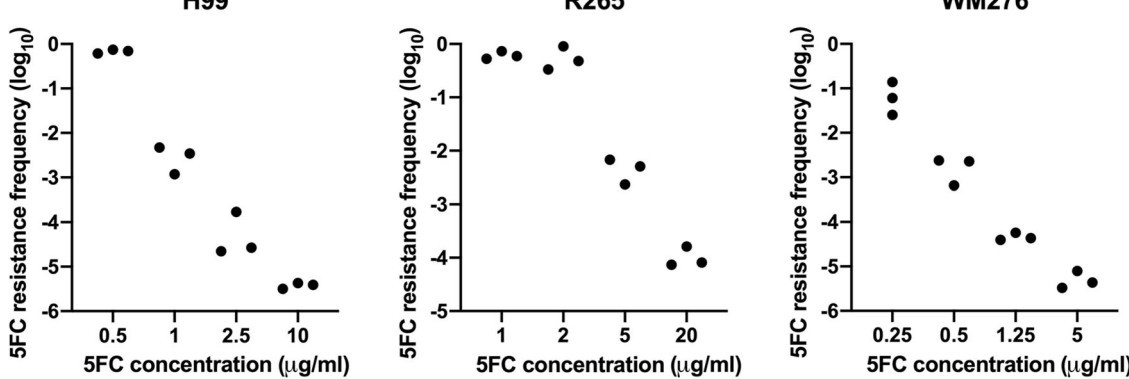

**Fig. 1 The frequency of 5-FC resistance varies according to 5-FC concentrations.** Cells from H99, R265, and WM276 were plated on YNB media supplemented with indicated amounts of 5-FC which were approximately corresponding to 20-, 40-, 100-, and 400-fold of 5-FC MIC of each tested strain. Plates were incubated at 30 °C for 7 days and photographed. The frequency of 5-FC resistance was calculated by dividing the number of the 5-FC resistant colonies by the total number of the input cells. The experiments were repeated three times.

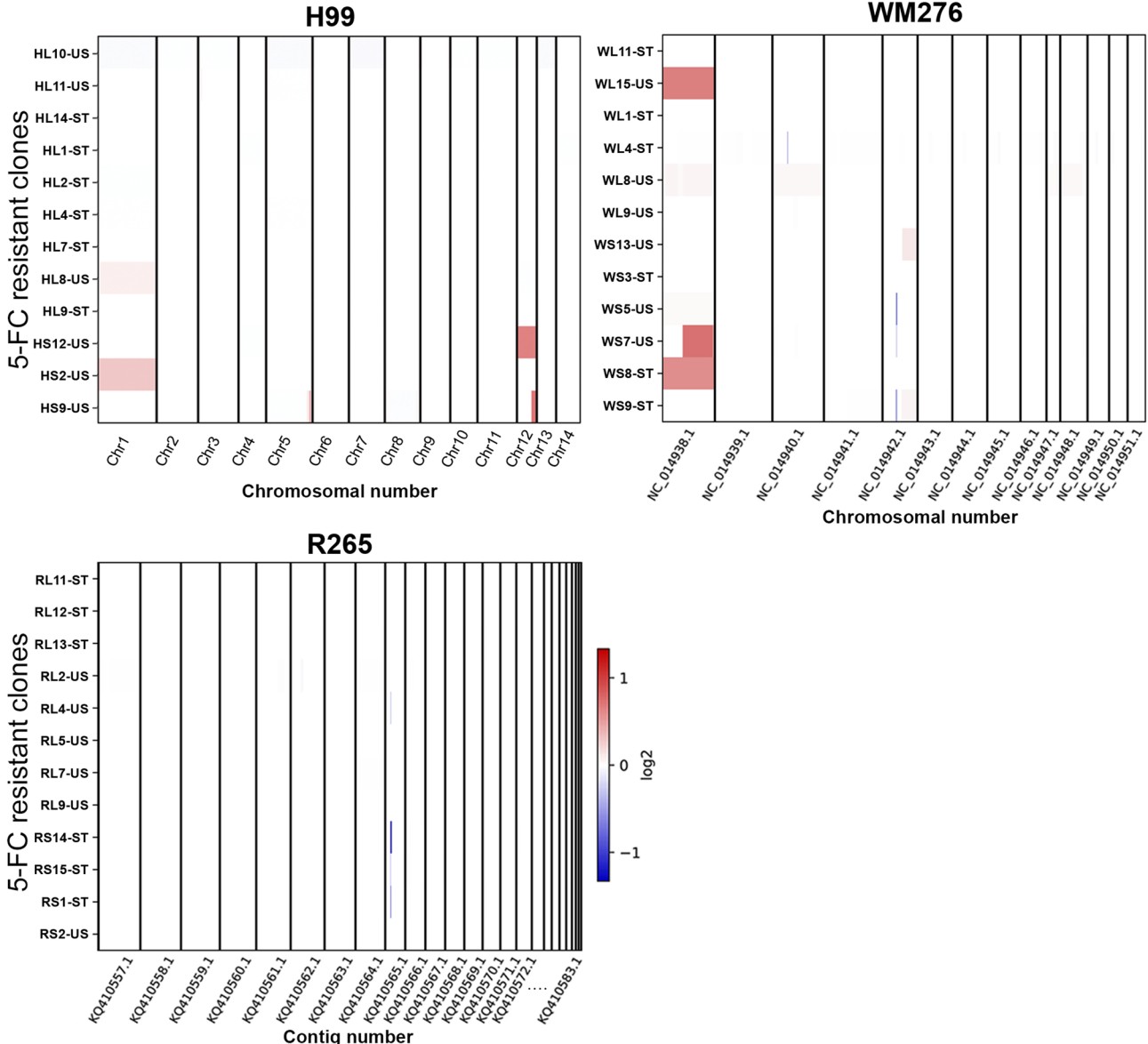

**Fig. 2 Chromosome copy number analysis.** The genomic sequences of twelve 5-FC resistant clones each from H99, R265, and WM276 were determined, and the copy number of each chromosome was calculated. The change of copy number is represented by the intensity of the color. The red color represents copy gain and blue color represents copy loss in units of log2 as shown by the color scale. The color bar represents the location of copy number change in the chromosome. The name of chromosome or contig is indicated at the bottom of each panel and the width of the column represents the size of chromosome or contig. The name of each clone is on the left side of each panel. The "L" and "S" in the name represent the size of the colony at the time of its isolation and ST and US in the suffix indicates the stability of each 5-FC resistant clone as either stable or unstable, respectively.

in the 5-FC resistant clones are exceedingly greater than the fluconazole heteroresistant clones in *Cryptococcus* species.

**Genomic sequencing of the 5-FC resistant clones**. It has been shown that the fluconazole heteroresistant clones derived from H99 and other clinical isolates are unstable aneuploid[25,26]. We performed whole-genome sequencing to determine if aneuploidy had occurred among the 5-FC resistant clones. We arbitrarily selected six each of stable and unstable 5-FC resistant clones derived from the three strains. The frequency of drug-resistant progeny at the end of daily transfer was greater than 90% in the stable clones and it was less than 7% in the unstable clones. We detected only a few clones had changes in the chromosome copy number. HL8, HS2, and HS12 were derived from H99 and had a slight increase in copy number of chromosome 1 (chr1) or chr12

and HS9 had an increase in a segment of chr12 (Fig. 2). Two WM276 derived clones, WL15, and WS8, had increased copy number of chr1 and WS7 had an increase in a segment of chr1. No clear increase in chromosome copy number was detected in clones derived from R265 except that two clones, RS14 and RS1, had a reduction of the copy number in a segment of contig KQ410565.1. Among the seven clones with a clear increase in chromosome copy number, only one clone, WS8, was stable while the rest of the clones lost their resistance during the transfers. These data suggest that the increase in chromosome copy number may contribute to the 5-FC resistance in some but not all of the 5-FC resistant clones.

**Chr1 duplication contributes to 5-FC resistance in H99.** Since the copy number of chr1 was elevated in HL8 and HS2, chr1

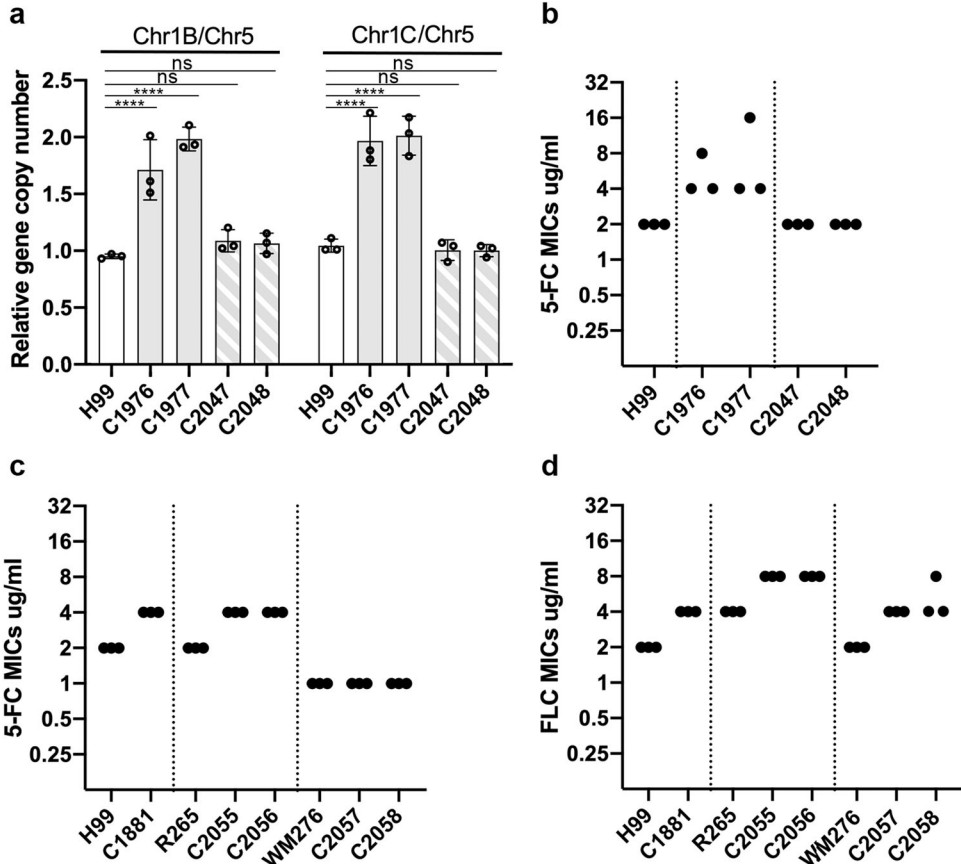

**Fig. 3 Chr1 duplication and extra copy of *AFR1* affects the 5-FC resistance levels. a** Relative copy number of genes resided on chr1. Probes specific for the genes resided on the left or right arms of chr1 (Chr1B or Chr1C) were chosen to assess the changes in chr1 copy number. The PCR results of the chr1-specific probes were compared to that of chr5, which served as unduplicated internal control in the indicated strains. Data are presented as mean values ± standard deviation from three biological repeats of each strain. Two-way ANOVA with Tukey's test for multiple comparisons (ns = not significant, **** = $p$ < 0.0001). **b** Broth 5-FC MIC of the strains containing duplicated chr1 and the strains after repeated transfers which lost the duplicated chr1. **c** Broth 5-FC MIC of the strains containing extra copies of *AFR1*. **d** Broth fluconazole MIC of the strains containing extra copies of *AFR1*. The MIC determination was repeated three times for each strain.

aneuploidy was presumed to have contributed to 5-FC resistance. This assumption was based on the observation that clones heteroresistant to 32 µg/ml fluconazole invariably contained duplicated chr1[25]. We generated two H99 derived heteroresistant clones, C1976 and C1977 (Supplementary Table 1), which were presumed to be disomic for chr1 and used them to determine the effect of chr1 duplication for 5-FC resistance. To assess the status of chr1 in C1976 and C1977, we analyzed the copy number of the genes located on each arm of chr1. The copy number of chr1 genes was 2-fold of the genes located on chr5, confirming duplication of chr1 in C1976 and C1977 (Fig. 3a). After repeated passages of C1976 and C1977 in drug-free YPD media, we isolated clone C2047 and C2048, respectively. The copy number of the chr1 genes in C2047 and C2048 was similar to the wildtype H99 (Fig. 3a) indicating that the extra copy of chr1 in C1976 and C1977 was lost during the repeated transfer. Notably, the 5-FC MIC of C1976 and C1977 by broth microtiter testing was 4–16 µg/ml compared to 2 µg/ml in H99 or the repeated passage strains, C2047 and C2048 (Fig. 3b). These results indicated that chr1 duplication contributed moderately to the 5-FC resistance though the mechanism remains unknown. Since chr1 contains *AFR1*, an ABC transporter gene important for fluconazole resistance[27], we inserted an extra copy of *AFR1* in the chr3 of the wild-type strain H99 to assess if two copies of *AFR1* gene could increase the 5-FC resistance levels. The 5-FC resistance levels in the resulting strain, C1881, were 2-fold higher than that in

H99 suggesting that *AFR1* in H99 plays a role in 5-FC resistance (Fig. 3c). Similarly, extra copies of *AFR1* in R265 derived clones, C2055 and C2056, exhibited slightly increased levels of resistance to 5-FC as well as fluconazole (Fig. 3c). In contrast, extra copies of *AFR1* in WM276 derived clones, C2057 and C2058, displayed increased levels of resistance only to fluconazole but not to 5-FC (Fig. 3c and d). These results indicated that the role of *AFR1* in 5-FC resistance is not consistent but strain dependent in *Cryptococcus* species complex.

**Identification of genetic variants in the genome of 5-FC resistant clones.** Since the majority of 5-FC resistant clones showed no clear variation of copy number in the genome, we performed variant analysis of the 36 sequenced genomes to explore the possible existence of any other alterations. Interestingly, an array of genetic variants was found among the resistant clones, including single nucleotide variant, insertion, deletion, and substitution (Supplementary Data 1). We focused our analysis only on the variants in the exon regions of the annotated genes. Several variants were found in the exon regions of *FCY1*, *FCY2*, and *FUR1* (Table 1). Two variants of *UXS1*, a recently identified gene important for 5-FC resistance[19], were found in WM276 derived clones, WL8 and WL9. Additionally, variants were detected in genes encoding a putative uridylate kinase, an epsilon DNA polymerase, a putative monosaccharide transporter,

**Table 1 Variants detected in the 5-FC resistance clones.**

| Clone name | Mutation Type | Nt change* | AA change | Gene symbol | functional annotation | Stability of 5-FC resistance | MIC (µg/ml) by MTS™ |
|---|---|---|---|---|---|---|---|
| H99 | | | | | | | 0.023–0.032 |
| HL1 | Deletion | c.414GA>G | | CNAG_02337 | Fur1 | stable | >32 |
| HL4 | Deletion | c.52AT>A | | CNAG_00613 | Fcy1 | stable | >32 |
| HL4 | Substitution | c.898 C>A | H300N | CNAG_04784 | Monosaccharide transporter | unstable | >32 |
| HL8 | Deletion | c.508GTTC>G | | CNAG_02337 | Fur1 | unstable | >32 |
| HL9 | Substitution | c.120G>T | L40F | CNAG_00613 | Fcy1 | stable | >32 |
| HS2 | Substitution | c.235G>A | G79S | CNAG_05935 | uridylate kinase | unstable | >32 |
| R265 | | | | | | | 0.064–0.094 |
| RL4 | Substitution | c.952G>A | A318T | CNBG_9478 | hypothetical protein | unstable | >32 |
| RL4 | Deletion | c.142GAAA>G | | CNBG_1368 | hypothetical | unstable | >32 |
| RL4, RL5 | Substitution | c.1549 G>A | A517T | CNBG_4985 | hypothetical protein | unstable | >32 |
| RL4, RS1, RS14 | Deletion | c.540CTTTTTG>C | | CNBG_4048 | Fur1 | un,st,st | >32 |
| RL9, RS1, RS14, RS15 | Deletion | c.75CTT>C | | CNBG_4048 | Fur2 | us,st,st,st | >32 |
| RL11 | Deletion | c.1260CTA>C | | CNBG_3227 | Fcy2 | stable | >32 |
| RL11 | Substitution | c.695T>C | M232T | CNBG_9545 | hypothetical protein | stable | >32 |
| RL12 | Substitution | c.470 C>T | A157V | CNBG_3227 | Fcy2 | stable | >32 |
| RS1 | Substitution | c.569 C>T | S190F | CNBG_2198 | cytoplasmic protein | stable | >32 |
| WM276 | | | | | | | 0.016–0.023 |
| WL1 | Substitution | c.260 G>A | G87D | CGB_E2660C | Fur1 | stable | >32 |
| WL8 | Deletion | c.829ATT>A | | CGB_G3260C | Uxs1 | unstable | >32 |
| WL9 | Deletion | c.538GTT>G | | CGB_G3260C | Uxs1 | unstable | >32 |
| WL11 | Deletion | c.460GAA>G | | CGB_A6420C | Fcy1 | stable | >32 |
| WL15 | Substitution | c.514G>T | V172L | CGB_F6195W | uridylate kinase | unstable | >32 |
| WS13 | Deletion | c.248GCT>G | | CGB_E1620C | transposable element | unstable | >32 |
| WS3 | Deletion | c.456ATC>A | | CGB_E2660C | Fur1 | stable | >32 |
| WS5 | Substitution | c.1522GG>T | G508C | CGB_F1400C | epsilon DNA polymerase | unstable | >32 |

*Only variants identified in the exon region are shown.

transposable elements, hypothetical proteins, and cytoplasmic proteins.

**Verification of the genes important for 5-FC resistance.** Before confirming the role of the variants containing genes in 5-FC resistance, we measured the levels of resistance to 5-FC in the 36 genome sequenced clones by MTS™ MIC test strips. All 36 clones appeared to be resistant to the highest concentration of 5-FC on the test strips, 32 µg/ml (Table 1). Since all of the sequenced clones contained more than one genomic variant (Supplementary Data 1), we constructed deletion mutants or generated the same variants postulated to be responsible for 5-FC resistance in the wild type strains.

First, we focused on the importance of *FCY1*, *FCY2*, and *FUR1* in 5-FC resistance and constructed deletion mutants of each gene in H99, R265, and WM276. Although *FCY1*, *FCY2*, and *FUR1* are known to play a critical role in 5-FC resistance in other fungi[6,15–17], their roles have not been unequivocally substantiated for cryptococcal 5-FC resistance[28] except that *FCY2* in R265 and *FUR1* in H99 have been shown to be important for 5-FC resistance[19,29]. Table 2 shows that the deletion mutant of each of the *FCY1*, *FCY2*, and *FUR1* gene derived from all three strains was resistant to >32 µg/ml 5-FC, confirming the major role of these three genes in cryptococcal susceptibility to 5-FC as in other fungi.

Variants were detected in two putative essential genes encoding uridylate kinase (*URA6*; CGB_F6195W) and epsilon DNA polymerase (*DPB2*; CGB_F1400C) in the WM276 derived clones WL15 and WS5 (Table 1). Ura6 catalyzes the seventh enzymatic step in the de novo biosynthesis of pyrimidines which converts uridine monophosphate into uridine-5′-diphosphate in *S. cerevisiae*[30]. *DBP2* encodes the second largest subunit of DNA polymerase II

and is required for the maintenance of chromosomal replication fidelity[31]. To determine the importance of *URA6* and *DBP2* in 5-FC resistance, we generated the same variants for the two putative essential genes in the wild type strain WM276 by molecular manipulations through homologous integration. The resulting strain C2080 containing *URA6*[V172L] variant displayed markedly higher MIC than the wild type (0.19–0.38 µg/ml vs. 0.016–0.023 µg/ml) (Table 2). This indicated that *URA6* contributes to 5-FC susceptibility which has not been implicated in other fungi. However, C2080 was not as resistant as WL15, the original carrier of *URA6*[V172L] variant (MIC > 32 mg/ml Table 1), suggesting that additional unverified changes might have rendered WL15 to be more resistant to 5-FC than C2080. In contrast, the *DPB2*[G508C] variant containing strain, C2171, had similar MIC as the wild-type WM276 (Table 2) indicating that the *DPB2*[G508C] variant was not involved in 5-FC resistance and other unverified changes in WS5 might be responsible for its resistance.

HL4 was a H99 derived clone that produced a large-sized resistant colony that contained two variants, a deletion variant in *FCY1* encoding region and a substitution variant, H300N, in the monosaccharide transporter CNAG_04784 (Table 1). Since importance of *FCY1* in 5-FC resistance has been confirmed, we obtained a CNAG_04784 deletion mutant, 32C12, from the Fungal Genetics Stock Center to determine the importance of CNAG_04784. The MIC of 32C12 was similar to the wild-type strain (Table 2) indicating that the H300N variant in CNAG_04784 of HL4 was unrelated to 5-FC resistance and suggested that the single nucleotide deletion in *FCY1* caused 5-FC resistance in HL4.

RS1 was a small resistant colony derived from R265 that contained a S190F variant in a cytoplasmic protein encoding

**Table 2 MIC determined by Liofilchem® MIC Test Strips™.**

| Strain | Description | 5-FC MIC (µg/ml) |
| --- | --- | --- |
| Strains containing gene deletion or substitution | | |
| H99 derived strains | | |
| H99 | wild type | 0.023–0.032 |
| C1918 | $fcy1\Delta$ | >32 |
| C1931 | $fcy2\Delta$ | >32 |
| C1920 | $fur1\Delta$ | >32 |
| 32C12 | $CNAG\_04784\Delta$ | 0.032–0.047 |
| 41H10 | $nrg1\Delta$ | 0.032–0.047 |
| R265 derived strains | | |
| R265 | wild type | 0.064–0.094 |
| C1960 | $fcy1\Delta$ | >32 |
| C1911 | $fcy2\Delta$ | >32 |
| C2178 | $fur1\Delta$ | >32 |
| C2172 | $CNBG\_2198\Delta$ | 0.064–0.125 |
| WM276 derived strains | | |
| WM276 | wild type | 0.016–0.023 |
| C1832 | $fcy1\Delta$ | >32 |
| C1834 | $fcy2\Delta$ | >32 |
| C1900 | $fur1\Delta$ | >32 |
| C2022 | 44 transfers of C1900 | >32 |
| C2080 | $URA6^{V172L}$ | 0.19/0.38 |
| C2171 | $DPB2^{G508C}$ | 0.012–0.016 |
| Strains related to suppressor | | |
| KN99 derived strains | | |
| KN99 | wild type | 0.032–0.047 |
| 13C2 | $uxs1\Delta$ | >32 |
| C1952 | 47 transfers of 13C2 | 0.064–0.094 |
| C2050 | $Knrg1$ in 13C2 | 0.064–0.125 |
| C2119 | $Kugd1$ in 13C2 | 0.047 |
| C2133 | $Knrg1$ in KN99 | 0.032–0.047 |
| C2168 | $Kugd1$ in KN99 | 0.032–0.047 |
| WM276 related strains | | |
| C2053 | $uxs1\Delta$ | >32 |
| C2074 | UXS1 complemented | 0.023 |
| C2104 | $Wugd1$ in $uxs1\Delta$ | 0.047–0.064 |
| C2072 | $Wnrg1$ in $uxs1\Delta$ | >32 |
| C2113 | $Xnrg1$ in $uxs1\Delta$ | >32 |
| C2061 | $Wugd1$ in WM276 | 0.012–0.023 |
| C2151 | Fcy2-mNG in WM276 | 0.032–0.047 |
| C2152 | Fcy2-mNG in $uxs1\Delta$ | >32 |
| C2153 | Fcy2-mNG in $uxs1\Delta + Wugd1$ | 0.047–0.064 |
| WL8 derived strains | | |
| WL8 | WM276 derived | >32 |
| C1933 | 51 transfers of WL8 | 0.016–0.023 |
| C2028 | $BCK1^{R1382*}$ in WL8 | >32 |
| C2143 | $Wugd1$ in WL8 | 0.016–0.023 |
| WL9 derived strains | | |
| WL9 | WM276 derived | >32 |
| C1943 | 51 transfers of WL9 | 0.032–0.047 |
| C2029 | $RAN1^{Y45*}$ in WL9 | >32 |
| C2030 | $Wugd1$ in WL9 | 0.032–0.047 |
| C2071 | $Wnrg1$ in WL9 | >32 |
| C2111 | $Xnrg1$ in WL9 | >32 |

gene, CNBG_2198 (Table 1). We deleted the CNBG_2198 gene in R265 and found the MIC of deletion mutant to be similar with the wild-type strain (Table 2) indicating that CNBG_2198 is unrelated to 5-FC resistance. We did not pursuit the variants found in the region encoding hypothetical protein in RL4 (CNBG_1368, CNBG_4985, and CNBG_9478) or RL11 (CNBG_9545) since these clones also contained variants of FUR1 or FCY2. We also did not further analyze the variant of the transposable element (CNAG_04784) in WS13 since multiple copies of the transposable element existed in the genome.

**Identification of the *uxs1* suppressors.** It was interesting to find that the UXS1 gene in the clones WL8 and WL9 derived from WM276 was partially deleted and yet their 5-FC resistance phenotype was unstable (Table 1). Similarly, the FUR1 gene was found partially deleted in both HL8 and RL4 clones derived from H99 and R265 respectively and their resistance phenotype was unstable. Because it is known that deletion of UXS1[19] or FUR1 causes resistance to 5-FC (Table 2), we thought that the instability of these clones could be due to the development of suppressors during repeated transfers in the nonselective medium. Since all four strains contained additional variants in the genome (Supplementary Data 1), we used the strains containing simple deletion of UXS1 in KN99α (strain 13C2) or FUR1 in WM276 (strain C1900) and repeated the transferring experiment to test the stability of their 5-FC resistance. Both 13C2 and C1900 were resistant to >32 µg/ml 5-FC but only the *uxs1* mutant (13C2) was unstable and lost resistance after repeated transfers (Table 2, 13C2 vs. C1952 and C1900 vs. C2022). Since the 5-FC resistance phenotype was consistently unstable in all three different types of *uxs1* deletion mutants, WL8, WL9, and 13C2, we focused our analysis on the *uxs1* mutants to identify the possible suppressors. We randomly selected and sequenced the genomes of five clones each derived from the three different *uxs1* strains that lost the resistance after repeated transfers. In all five clones derived from 13C2, we identified a same deletion variant at the C-terminal end of CNAG_05222 which encodes a transcriptional regulator Nrg1 (Supplementary Data 2). Additionally, in all the five clones derived from WL9, we identified a missense variant of CGB_D0330C which encodes a UDP-glucose 6-dehydrogenase Ugd1, and a nonsense variant of CGB_F0330C which encodes a RAN protein kinase Ran1. While we found no commonly shared variants among the five WL8 derived clones, a missense variant of UGD1 in four and a nonsense variant of CGB_I2500W which encodes a mitogen-activated protein kinase kinase kinase Bck1 in three among the WL8 derived clones (Supplementary Data 2).

To verify the importance of the identified variants in suppression of the *uxs1*-induced 5-FC resistance, we regenerated the same type of variants in the original *uxs1* mutants, 13C2, WL8, and WL9 by molecular manipulations. When the native NRG1 gene in 13C2 was replaced by the NRG1 deletion variant, $Knrg1$, the resulting clone lost its resistance (Table 2, C2050). This result indicated that the nrg1 variant could function as a suppressor for $uxs1\Delta$ in KN99α background. When the $UGD1^{G19A}$ variant, $Wugd1$, was introduced into WL9, the resistance phenotype was suppressed (Table 2, C2030) but the $RAN1^{Y45*}$ variant failed to suppress the phenotype (Table 2, C2029) suggesting that only the $UGD1^{G19A}$ variant was responsible for suppression of the resistance in WL9 derived clones. In contrast, the $BCK1^{R1382*}$ variant failed to suppress the 5-FC resistance phenotype in WL8 (Table 2, C2028). Although we did not examine the function of $UGD1^{I54F}$ variant detected in the WL8 derived clones, the $UGD1^{G19A}$ variant found in WL9 derived clones was able to suppress the 5-FC resistance in WL8 (Table 2, C2143). We also generated a *uxs1* deletion mutant, C2053, in WM276 background. C2053 was resistant to >32 µg/ml 5-FC and when the $uxs1\Delta$ mutation was complemented, the resulting strain lost its resistance (Table 2, C2074). Furthermore, when the $UGD1^{G19A}$ variant was introduced into C2053, the resulting strain lost its resistance (Table 2, C2104). These data indicated that the $UGD1^{G19A}$ variant could function as a suppressor in all three types of *usx1* deletion mutants in WM276 background.

Although WM276 (*C. gattii*) and KN99α (*C. neoformans*) belong to closely related species, numerous differences were found in the sequences of UGD1 and NRG1 between the two strains. To determine if the identified variants could function as suppressors in

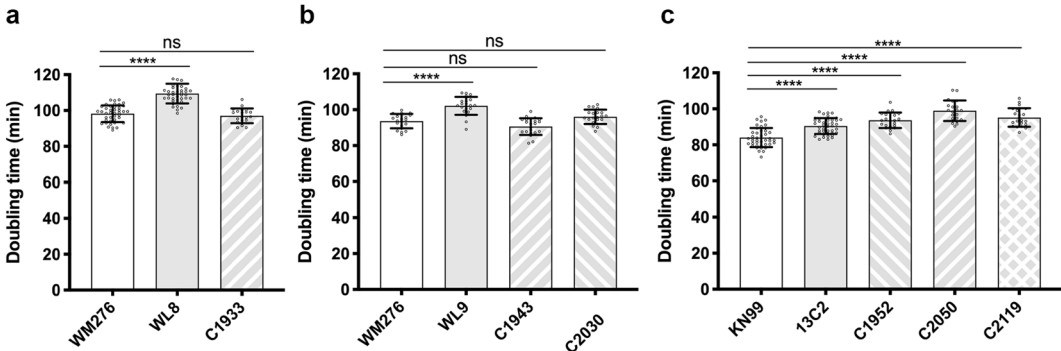

**Fig. 4 Growth rate is different between the _uxs1_ suppressor strains in _C. neoformans_ and _C. gattii_.** The doubling time of indicated strain is determined using a plate reader. **a** C1933 is a 5-FC sensitive strain derived from repeated transfers of WL8. **b** C1943 is a 5-FC sensitive strain derived from repeated transfers of WL9. C2030 is a strain containing _UGD1_$^{G19A}$ that was manually introduced into WL9. **c** C1952 is a 5-FC sensitive strain derived from repeated transfers of 13C2. C2050 and C2119 are suppressor strains obtained by molecular manipulations which contained _Knrg1_ and _UGD_$^{G19A}$, respectively. The experiments were repeated at least three times ($n \geq 21$) and data are presented as mean values ± standard deviation. One-way ANOVA with Tukey's test for multiple comparisons (ns = not significant, and ****$p < 0.0001$).

both species, we generated the same type of _UGD1_ and _NRG1_ variants based on the sequences of WM276 and KN99α respectively. The _UGD1_$^{G19A}$ variant designed according to KN99α, _Kugd1_, was able to suppress the 5-FC resistance in the 13C2, a _uxs1Δ_ mutant of KN99α (Table 2, C2119). This indicates that _UGD1_$^{G19A}$ variant originally detected in WM276 (_C. gattii_) derived clones could also function as a suppressor of _uxs1_ in _C. neoformans_. However, the partially deleted _NRG1_ variant designed according to WM276, _Wnrg1_, failed to suppress the 5-FC resistance in _uxs1Δ_, C2053, and WL9 derived from WM276 (Table 2, C2072 and C2071). Since the N-terminal sequence of Nrg1 is conserved between WM276 and KN99α but the C-terminal sequence is diverged, we constructed a mosaic _nrg1_ which contained the wild type _NRG1_ of WM276 in the 5′-end and the _NRG1_ deletion variant of KN99α in the 3′-end. The resulting construct, _Xnrg1_, still failed to suppress the 5-FC resistance phenotype of _uxs1_ mutants in WM276 background (Table 2, C2113 and C2111) indicating that the _NRG1_ deletion variant only functioned as a suppressor in _C. neoformans_ but not in _C. gattii_.

**The growth rate is different among different _uxs1_ suppressor strains.** We suspected that the growth rate of WL8, WL9, and 13C2 had been retarded due to the _uxs1_ mutations and cells with faster growth rate could have been selected out through repeated passages in drug-free media and became dominant at the end of the experiments. Figure 4 shows that the doubling time of WL8 and WL9 was significantly longer than that of WM276 which indicates that the growth rate had been retarded in the WM276 _uxs1_ mutants. In contrast, the doubling time of clones derived from WL8 and WL9 after repeated transfers, C1933 and C1943, respectively, was significantly shorter than the parental strains and was similar to the wild type WM276 (Fig. 4a and b). Furthermore, introducing _UGD1_$^{G19A}$ into WL9 also produced a strain with similar doubling time as the wild type (Fig. 4b, C2030). These data indicated that repeated transfer of the _C. gattii_ _uxs1_ mutants in the drug-free medium was selective for cells with faster growth rate. Surprisingly, although the doubling time of 13C2 was significantly longer than KN99α, the doubling time of C1952 generated from the repeated transfers of 13C2 was similar to 13C2 and was significantly longer than KN99α (Fig. 4c). Therefore, repeated transfer of KN99α _uxs1Δ_ mutant in the rich medium did not select for cells with growth rate advantage. Furthermore, manual introduction of the _Knrg1_ or _UGD1_$^{G19A}$ variant into 13C2 did not improve its growth rate (Fig. 4c, C2050 and C2119). These results indicated that in _C. neoformans_, the

_uxs1Δ_ suppressors fail to improve the growth rate defect of the _uxs1Δ_ mutant.

**5-FC susceptibility is correlated with the UDP-GlcUA levels in the cells.** It has been shown that the _uxs1Δ_ mutant of _C. neoformans_ accumulates high amounts of nucleotide sugar UDP-GlcUA[32] and accumulation of UDP-GlcUA has been proposed to suppress the toxicity of 5-FC[19]. We measured the levels of nucleotide sugars in the cells to determine if the suppressors of the _uxs1_ mutants could modulate the levels of UDP-GlcU. Figure 5a shows that UDP-GlcUA concentration was significantly higher in the _uxs1_ deletion mutants C2053, WL8, and WL9 derived from WM276 indicating that _C. gattii_ _uxs1Δ_ mutants accumulate high levels of UDP-GlcUA as is the case in _C. neoformans_[32]. Introduction of the _UGD1_$^{G19A}$ variant into either of the three _uxs1_ deletion mutants, C2053, WL8, and WL9, significantly reduced the concentration of UDP-GlcUA in the cells (Fig. 5a. C2104, C2143, and C2130, respectively). Similarly, the _uxs1Δ_ mutant of KN99α, 13C2, accumulated significantly higher amounts of UDP-GlcUA compared to that of the wild type KN99α (Fig. 5b). Also, C2050 and C2119, which contained either the _nrg1_ deletion variant or the _UGD1_$^{G19A}$ variant in 13C2, synthesized similar amounts of UDP-GlcUA as the wild type (Fig. 5b). These results indicated that the _uxs1Δ_ mutants of both cryptococcal species accumulate high levels of UDP-GlcUA and the accumulation is diminished in the suppressor containing strains.

**The Ugd1 function or the _UGD1_ expression levels is reduced in suppressor containing strains.** We have identified two types of genomic suppressors in _uxs1Δ_ mutants. One is mutation in _UGD1_ and the other is genetic alteration in _NRG1_. _UGD1_ and _UXS1_ are functionally related genes for capsule synthesis in _C. neoformans_[32–35]. _UGD1_ encodes a UDP-glucose 6-dehydrogenase which converts UDP-glucose to UDP-glucuronic acid[32]. _UXS1_ is downstream of _UGD1_ and encodes an enzyme that converts UDP-GlcUA to UDP-xylose[35]. It is known that loss-of-function mutations in upstream components of a pathway can prevent the accumulation of the compound caused by mutation of the downstream gene and form suppression. For example, in _S. cerevisiae_, deletion of _ADE13_ leads to accumulation of a toxic metabolite (S)-2-[5-Amino-1-(5-phospho-D-ribosyl)-imidazole-4-carboxamido]succinate and causes a growth defect, which can be suppressed by loss of upstream pathway components[36]. Unlike the loss-of-function mutations in Ade13 suppressors, the

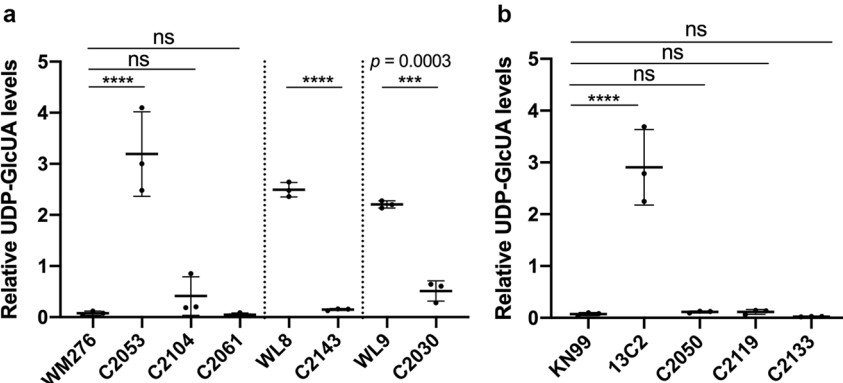

**Fig. 5 The levels of UDP-glucuronic acid are high in the *uxs1* mutants and low in the suppressor strains.** Nucleotide sugars were analyzed as described under "Methods". Relative amounts of UDP-glucuronic acid were determined in strains derived from WM276 (**a**) and KN99α (**b**). The experiments were repeated three times and data are presented as mean values ± standard deviation. One-way ANOVA with Tukey's test for multiple comparisons (ns = not significant, ***$p < 0.0002$, and ****$p < 0.0001$).

$UGD^{G19A}$ variant is functional in terms of producing UDP-GlcUA (Fig. 5). However, the activity of mutant protein may have been suffered since the amounts of UDP-GlcUA in the suppressor strain of the *uxs1Δ* mutants was not significantly higher than that of the wild type. In addition, while the *ugd1Δ* mutant was acapsular, the $UGD^{G19A}$ containing strains were encapsulated in both KN99α and WM276 backgrounds (Fig. 6a). But the capsule size in the $UGD^{G19A}$ containing strains was slightly smaller than that of the wild type in both KN99 and WM276 suggesting that $UGD^{G19A}$ variant may have lost some of its functions (Fig. 6c). The second type of *uxs1* suppressor, *NRG1*, was found partially deleted in KN99α. *NRG1* encodes a transcription factor that has been shown to regulate the expression of *UGD1*[37]. It is possible that the action of suppression by *Knrg1* is through down-regulation of *UGD1* since we found the expression levels of *UGD1* were reduced in *Knrg1* containing strains compared to the wild type (Fig. 6d). Therefore, the Ugd1 function or the *UGD1* expression was reduced in the suppressor containing strains which may result in the decrease of UDP-GlcUA accumulation.

**UXS1 regulates intracellular 5-FC accumulation.** It was not clear how accumulation of UDP-GlcUA could alter cryptoccal susceptibility to 5-FC. One can speculate that a high concentration of UDP-GlcUA could affect the cellular intake of 5-FC. We determined the accumulation of tritium-labeled 5-FC in various strains through time course experiment. After incubation of cells in the presence of [³H]-5-FC for 20 min or longer, all three types of WM276 *uxs1* deletion mutants accumulated significantly lower amount of [³H]-5-FC compared to the wild type (Fig. 7 C2053, WL8, and WL9 vs. WM276). When the $UGD1^{G19A}$ variant was introduced into the genome of each *uxs1* deletion mutant, the accumulation of [³H]-5-FC increased significantly compared to that of each deletion strain (Fig. 7, C2104, C2143, and C2030). Similarly, the *uxs1Δ* mutant of KN99α, 13C2, accumulated significantly lower levels of [³H]-5-FC than the wild-type strain. When either the *nrg1* deletion variant or the $UGD1^{G19A}$ variant was introduced into 13C2, the accumulation of [³H]-5-FC was significantly increased compared to 13C2 (Fig. 7d, C2050 and C2119). These data indicate that *UXS1* and its suppressors can regulate the intracellular accumulation of 5-FC both in *C. neoformans* and *C. gattii*.

**UXS1 modulates the expression levels of the purine-cytosine permease.** Since 5-FC enters the cells primarily via the purine-cytosine permease Fcy2 in many fungi and the cryptococcal *fcy2Δ* mutants were resistant to 5-FC (Table 2), it was possible that the

observed alteration of the 5-FC accumulation in *uxs1* deletants was caused by modification of the cellular localization or expression levels of Fcy2. We tagged Fcy2 at its N-terminus with mNG and FLAG at the native *FCY2* locus in WM276, *uxs1Δ*, and $UGD1^{G19A}$ suppressor strain respectively. The tagged strains, C2151, C2152, and C2153, had the same 5-FC resistance levels as their parental strains (Table 2) suggesting that N-terminal tagging of Fcy2 did not affect the function of Fcy2. Microscopic examinations revealed that the Fcy2 was mainly localized to the cell membranes a well as in the structures reminiscent of large vesicles in all three tagged strains (Fig. 8a). Additionally, the fluorescence intensity was slightly lower in the *uxs1Δ* mutant compared to that of the wild type and the strain of *uxs1D* containing $UGD1^{G19A}$ (Fig. 8b). Western blot analysis confirmed that Fcy2 protein levels were significantly lower in the *uxs1Δ* mutant than the wild type and $UGD^{G19A}$ strains (Fig. 8c). These results indicated that the status of *UXS1* and the presence of its suppressors regulated the accumulation of 5-FC inside the cells via modulation of the Fcy2 expression levels.

## Discussion

Our study reveals that 5-FC induces various types of genetic modifications in both *C. neoformans* and *C. gattii*. Genetic modifications appeared to have been random event since various genomic variants were detected in those areas related as well as unrelated to 5-FC resistance. The high frequency of resistant mutants reported here has biological relevance since it implicates 5-FC to be mutagenic. The frequency of resistance far exceeds the classical spontaneous mutation frequency in eukaryotes and hence, it is unlikely that the resistant clones represent unusually large number of pre-existing spontaneous mutants. It is known that 5-FC is converted to 5-fluorouracil inside the cells. 5-fluorouracil inhibits thymidylate synthase and thymidine starvation leads to single- and double-strand DNA breaks. Thymidine deprivation also leads to incorporation of uracil into DNA in place of thymidine and causes mutations in other eucaryotes[38–40]. 5-fluorouracil is massively incorporated into ribonucleic acid causing abnormal rRNA assembly, tRNA misfolding, and mRNA misreading[41–44]. It is possible that at moderate concentrations, 5-FC may not sufficiently inhibit cell growth and the metabolized 5-FC can cause various modifications including chromosome aneuploidy. Subsequently, cells containing modifications in genes relevant for 5-FC resistance are selected and form resistant colonies at high frequency. At high concentrations, however, 5-FC appears to inhibit the growth effectively and select out spontaneous mutants with genetic alterations in the genes relevant for 5-FC resistance. It is known that spontaneous mutation frequency can vary widely

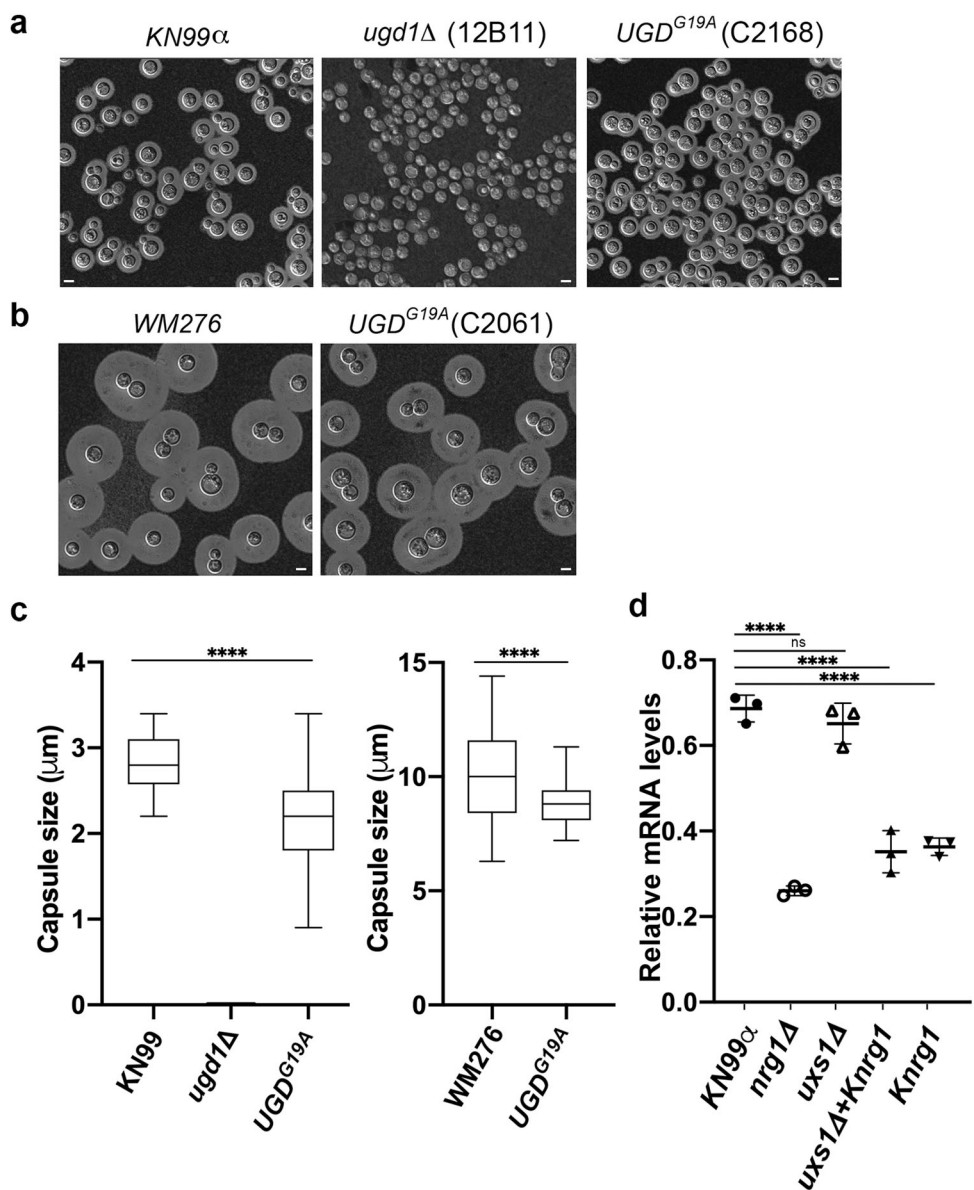

**Fig. 6 The Ugd1 function or the *UGD1* expression levels is affected in suppressor containing strains. a–c** Capsule size is affected by the *UGD1^G19A* mutation. Cells were stained by India ink and visualized by microscopy. The experiments were repeated 4 times independently with similar results. The capsule sizes of more than 65 cells were measured from 5 different fields for each indicated strain. Capsular thickness was defined as the distance between the visualized cell wall and the edge of the capsule. The box-and-whiskers plot is used to show median, quartiles (boxes), and range (whiskers). Two-tailed, unpaired t-test (****$p < 0.0001$). Bar = 5 μm. **d** *UGD1* expression is downregulated in the *Knrg1* containing strains. RNA was isolated and the expression levels of *UGD1* were determined by qRT-PCR, normalized to the actin gene. The experiments were repeated three times. One-way ANOVA with Tukey's test for multiple comparisons (ns = not significant and ****$p < 0.0001$).

depending on the strains within the same species. For example, mutation frequency to rifampicin resistance in *Helicobacter pylori* varied between $3 \times 10^{-5}$ and $4 \times 10^{-8}$ among tested strains[45]. In this study we identified at least 5 genes that contribute to 5-FC resistance. At concentrations close to 400-fold of MIC, both *C. neoformans* (H99, 10 μg/ml) and *C. gattii* (WM276, 5 μg/ml) developed 5-FC resistant colonies at the frequency close to $1 \times 10^{-6}$ which is within the ranges of classical spontaneous mutation frequency. It is difficult to extrapolate our in vitro results in the clinical settings since the 5-FC dosage recommended for therapy is based on the body weight of patients while our study was conducted on agar plates at 30 °C. Although it has been recommended that the 5-FC dose in patients be adjusted to maintain the serum 5-FC levels at 50 to 100 μg/ml[46], possibility still exists that 5-FC can

induce cryptococcal genomic changes in the host during long term treatment and cause therapeutical failure.

In understanding the regulation of 5-FC drug susceptibility in cryptococci, we identified several factors that contribute to the changes in susceptibility including mutations in *FCY2, FCY1, FUR1, UXS1,* and *URA6* genes in addition to several unverified variants that remains to be explored; developing of suppressors; chromosome aneuploidy and the regulation of the gene on aneuploid chromosome. Aneuploidy is known to be associated with an adaptation process in pathogenic yeasts to fluconazole stress[25,47,48]. The most prevalent aneuploidy found in fluconazole heteroresistant pathogenic yeasts is the chromosome that harbors the genes encoding the target of azoles, the major efflux pump or transcription regulator of drug efflux pumps[25,47]. Our results

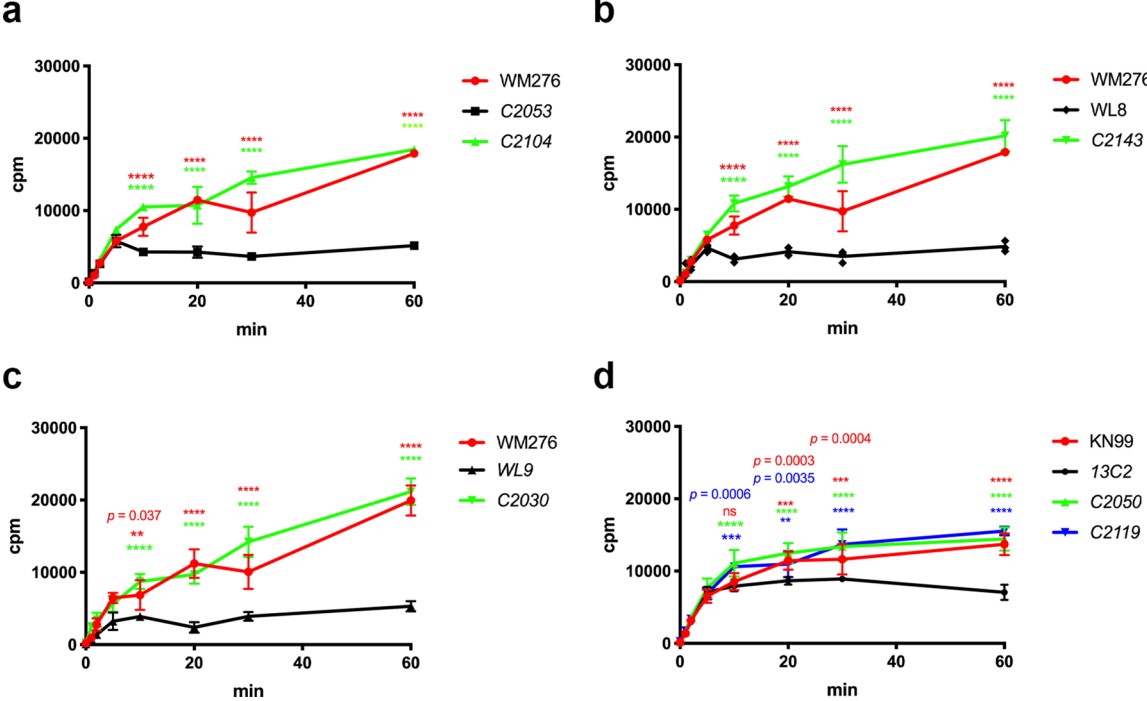

**Fig. 7 The accumulation of [³H] 5-FC is low in the *uxs1* mutants and high in the suppressor strains.** Cells were incubated with [³H]-flucytosine and the accumulation of [³H]-flucytosine was determined at indicated time. The experiments were repeated three times and data are presented as mean values ± standard deviation. Each panel contains the wild type, the *uxs1* mutant and the suppressor strains derived from WM276 (**a**), WL8 (**b**), WL9 (**c**) and KN99α (**d**). Two-way ANOVA with Tukey's multiple comparisons test was used to compare the differences at the indicated time point between the *uxs1* mutant and the wild type or between the *uxs1* mutant and the suppressor strain (ns = not significant, $**p < 0.0021$, $***p < 0.0002$, and $****p < 0.0001$).

indicated that chr1 aneuploidy was also beneficial for cryptococcal cells in responding to 5-FC stress. Interestingly, an extra copy of the ABC transporter gene, *AFR1*, slightly increases the drug resistance levels in H99. Although Afr1 is known as a primary azole efflux pump and deletion of *AFR1* has no clear impact on the level of 5-FC resistance[49], Afr1 might have a supplementary minor role in transporting other xenobiotics such as 5-FC in combination with other transporters. Thus, extra copy of *AFR1* could contribute modestly to the 5-FC resistance in H99. It is also interesting that extra copies of *AFR1* slightly increased the resistance levels of 5-FC in R265 but not in WM276. We observed several other differences among the three strains analyzed. For instance, 5-FC MIC was higher in R265 compared to H99 and WM276 and R265 had a higher frequency of 5-FC resistant colonies than H99 and WM276 at similar folds of 5-FC MIC levels. Moreover, increased chromosome copy number was not found among the sequenced drug-resistant clones derived from R265 and the growth rate of the suppressor strains was improved in WM276 but not in KN99α background. Additionally, *NRG1* variant was able to suppress 5-FC resistance phenotype of the *uxs1* mutant in *C. neoformans*, KN99α but not in *C. gattii*, WM276. These results indicate that the genetic wiring for 5-FC resistance is divergent between different strains of the cryptococcal species complex.

Isolation of suppressors for 5-FC resistant phenotype was serendipitous during repeated transfers of the *usx1* mutants in nonselective medium. The growth rate of the *uxs1* mutants containing *UGD1^G19A* suppressor was improved in the WM276 background, which could have increased the chance of isolating suppressor strains. It is not clear how the *UGD1^G19A* suppressor influences the growth in WM276 background. Since the presence of the suppressors reduced the accumulation of UDP-GlcUA, it is possible that such a reduction could restore the growth defect of *uxs1Δ* mutant in WM276. In contrast, the growth rate was not

improved in the suppressors containing KN99α *uxs1Δ* mutants. It is unclear how repeated transfers of the KN99α *uxs1Δ* strains in drug-free media resulted in the selection for the suppressor containing cells with no growth rate advantage. One possible scenario can be that deletion of *UXS1* in KN99α may reduce life span of the cells and the suppressor containing cells could overcome the deficiency. As a consequence, the *uxs1* mutant cells could have been outnumbered by the suppressor containing cells during repeated transfers.

It is not clear how exactly UDP-GlcUA accumulation might impact permease expression. It is possible that high levels of UDP-GlcUA might affect the expression of permease by altering the expression of transcription factor(s). However, we have failed to identify a probable transcription factor gene through analysis of transcriptome profile using the RNAs isolated from the wild-type, the WM276 *uxs1Δ* mutant, and suppressor containing strains in the *uxs1Δ* and wild-type background. It is also possible that accumulation of UDP-GlcUA might affect the Fcy2 levels at the translational or post-translational steps. The detailed mechanism(s) of how UDP-GlcUA might affect permease expression remains to be elucidated.

## Methods

**Strains and culture conditions.** Strains relevant to the study are listed in Supplementary Table 1. KN99α is derived from H99[50]. Strains were stored in 25% glycerol stocks at −80 °C until use and were maintained on YPD (1% yeast extract, 2% peptone, 2% glucose) agar plates at 30 °C for routine cultures. For the 5-FC resistant clones, the cultures were revived on YNB (yeast nitrogen base without amino acids) agar medium supplemented with 5-FC at the concentrations from which the clones were originally isolated. For capsule detection, cells were grown in Dulbecco's modified Eagle's medium (Gibco Lab-oratories, GrandIsland, NY) with 25 mM glucose (4.5 g/l) supplemented with 22 mM NaHCO₃, 25 mM NaMOPS pH 7.3 with 5% CO₂ at 37 °C for 48 h.

**Determination of MIC for antifungal antibiotics.** Unless specified, the 5-FC MICs were determined by using the Liofilchem® MTS™ (MIC Test Strips) (Liofilchem,

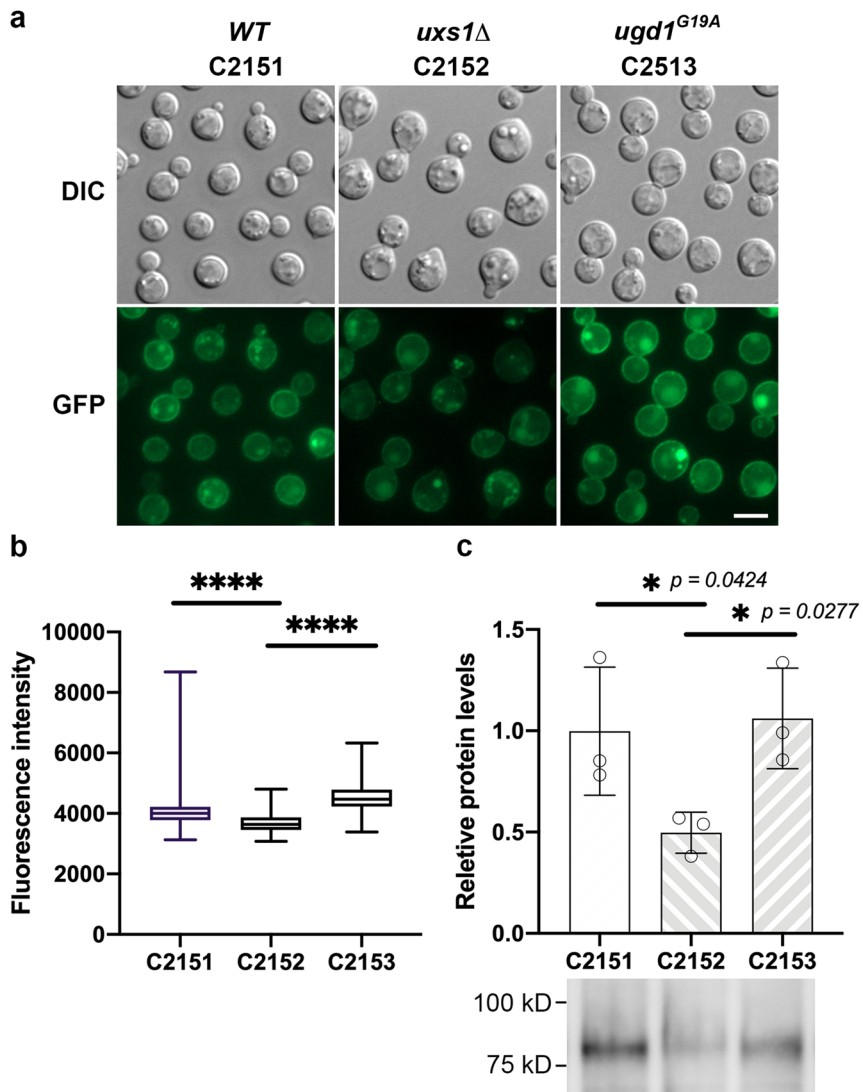

**Fig. 8 Fcy2 protein levels are low in *uxs1* mutants and high in the suppressor strains. a** The localization of Fcy2 in the indicated strains was visualized by microscopy and photographed. Bar = 5 μm. **b** The mean fluorescence intensity of each strain from (**a**) were quantified from three separate images. The box-and-whiskers plot is used to show median, quartiles (boxes), and range (whiskers). One-way ANOVA with Tukey's test for multiple comparisons (****$p < 0.0001$). **c** The relative amounts of the FLAG tagged Fcy2 was analyzed by western blot (bottom) and the expression levels of Fcy2 were quantified by measuring the band intensity in the western blots, normalized to that of the total protein from stain-free blot and compared to the expression levels in C2151. For the full scan blots, see the Source data file. Data are presented as mean values ± standard deviation from three biological repeats. One-way ANOVA with Tukey's test for multiple comparisons (*$p < 0.0332$ and ****$p < 0.0001$).

Waltham, MA) with modifications. Approximately $2 \times 10^6$ cells were plated on YNB agar plates followed by application of test strips. The plates were incubated at 30 °C and photographed after 72 h. We noted that the MIC results from Liofilchem® MTS™ were lower than the previously published results which was based on AB Biodisk Etest strips[29]. When MIC levels of mutant strains were close to the wild type, we used the broth MIC method recommended by CLSI M27-A3[51] to determine the MIC levels. Briefly, 5-FC or fluconazole was dispensed in 96-well microtiter plates with 2-fold serial dilutions of drug concentrations ranging from 32 μg/ml to 0.0625 μg/ml in MOPS-buffered RPMI 1640 and incubated at 37 °C for 72 h.

**Determination of the 5-FC resistance frequency**. Three days old cells grown on YPD plate were harvested and plated on YNB media supplemented with 5-FC at various concentrations which corresponded to approximately 20-, 40-, 100-, and 400-fold of 5-FC MIC of each tested strain. Plates were incubated at 30 °C for 7 days and photographed with ChemiDoc MP using Image Lab software (Bio-Rad, Hercules, CA). Because the size of some of the 5-FC resistant colonies varied, we used CellProfiler 3.1.5 software[52] to unbiasedly determine the number of 5-FC resistant colonies in the photograph. We considered the colonies with minimal diameter greater than 8 pixels as the 5-FC resistant clones in all concentrations of 5-FC. The frequency of 5-FC resistance was calculated by dividing the number of the resistant colonies obtained from CellProfiler by the total number of the input cells.

**Stability test of 5-FC resistant clones**. The clones used for stability test were originally isolated from plates containing 0.5 μg/ml 5-FC for H99 (approximately 20-fold of MIC) as well as WM276 (approximately 40-fold of MIC) and 5 μg/ml 5-FC for R265 (~100-fold MIC). The 5-FC resistant clones were grown in 2 ml of YPD liquid media at a 30 °C shaker incubator and 20 μl of the culture was transferred to fresh YPD media daily for up to 51 days. At each indicated period during the daily transfer, cells from the YPD liquid culture were diluted and plated out on 5-FC plates and the percentage of the 5-FC resistant colonies was determined. For simplicity, the resistance frequency at or greater than 100% is expressed as 100%.

**Whole-genome sequencing**. The 5-FC resistant clones were individually expanded by culturing for 3 days at 30 °C on YNB plate supplemented with 5-FC at the concentrations in which the clones were originally isolated. For the 5-FC susceptible clones, individual clones were grown on YPD plate at 30 °C for 2 days. Cells were harvested from the plates and lyophilized. Genomic DNA was isolated using the cetyltrimethylammonium bromide (CTAB) extraction method[53] with modifications. Paired-end libraries (150 bp) were prepared and sequenced using an Illumina HiSeq 4000 platform (Novogene, Davis, CA). The analysis was generated using Partek® Flow® software, version 7.0, 2017 (Partek Inc., St. Louis, MO). Illumina reads were trimmed from both end with minimal quality level at 38 and aligned to the reference genome of H99 (CNA3), R265 (V2) and WM276

(ASM18594v1), respectively, using BWA. Genome-wide copy number was determined by CNVkit with WGS method and the results were used to generate heatmap with CNVkit[54]. SNP and indel calling were performed using FreeBayes variant caller with haploid setting. The data was filtered by allele frequency greater than 4.5 and read depth greater than 5. FungiDB was also used to determine putative function and orthology of the genes which contained features called variants in the dataset[55].

**Gene manipulation**. Standard molecular cloning combined with overlap PCR and In-Fusion® HD Cloning kit (Clontech, Mountain View, CA) were used to build the desired constructs. To generate the construct of each variant, primers carrying desired modifications (Supplementary Table 2) were used for PCR amplification and cloned into pCR2.1 vector (Invitrogen, Carlsbad, CA) along with indicated selectable markers. To make *Xnrg1*, the 3′-end of *NRG1* in *Wnrg1* was digested with *Nhe*I and *Bsu*36I and replaced with the *Nhe*I/*Bsu*36I fragment derived from *Knrg1*. Gene deletion, complementation, tagging, and allele replacement were obtained by biolistic transformation. Homologous integration was confirmed by PCR and Southern blot analysis.

The *AFR1* gene was amplified by PCR from H99, cloned into pCR2.1 along with the *NEO* selectable maker and inserted in the intergenic region between CNAG_03012 and CNAG_03013 of Chr3[25] to generate pYCC1200. To construct the strains containing an extra copy of *AFR1* in chr3 of H99, pYCC1200 was transformed into H99 and the corrected clones were identified by PCR and Southern blot analysis. pYCC1200 was also transformed into R265 and WM276 to generate strains contained extra copies of *AFR1* at ectopic sites.

**Determination of growth rate**. The growth curves for each strain were obtained by using 48 wells microtiter plates (Costa 3548, Mesa, CA) with Synergy H1 plate reader (Biotek, Winooski, VT). Cells were grown in 5 ml YPD for 22–24 h at 30 °C with shaking. The culture was diluted to OD = 0.05 in YPD and 500 µl of cells was applied to each well. For each strain, 7 wells were used and one well containing only YPD was used as a control. Plate was sealed with Breathe-Easy film (Diversified Biotch, Boston, USA) and placed in the plate reader at 30 °C. The program setting was double orbital, 2 mm fast with 548 cpm and the reading was taken at 15 min intervals for 24 h. The growth rate was obtained from at least three independent experiments. The Max V was obtained from log phase using Gen5 Data Analysis Software (Biotek, Winooski, VT) and the doubling time was calculated from Max V.

**Isolation of nucleotide sugar**. Nucleotide sugars were analyzed from total protein extracts[32]. Samples were analyzed by a Thermo Scientific Vanquish UPLC coupled with a Q-Exactive high resolution accurate-mass spectrometer (HRAM-MS, Thermo Scientific, Waltham, MA) with heated electrospray ionization (HESI-II) in negative ion mode. 50 µl of 2 µg/µl protein samples were added to 200 µl internal standard solution containing 100 ng/ml of $^{13}C_6$-Saccharin in acetonitrile, vortexed for 5 min, and then centrifuged at 4 °C, 20,000 × g for 15 min. 200 µl supernatant was transferred into a LC-MS vial, and mixed with 200 µL of $H_2O$ for analysis. An Agilent Infinitylab Poroshell 120 EC-C18 column (2.1 mm × 100 mm, 1.9 µm) was used. Column was maintained at 40 °C and samples were kept in the autosampler at 4 °C. The injection volume was 3 µl. Chromatographic conditions were as follows: Solvent A: 5% MeOH, 3.5 mM dibutylamine, pH 5.5 (adjust with AcOH), Solvent B: 62% ACN, 33% IPA, 5% H2O, 3.5 mM dibutylamine, pH 5.5 (adjust with AcOH). The flow rate was 225 µl/min, and the gradient was 10% B at 0 min for 0.15 min, increased to 100% B at 3 min, remained at 100% B until 4.5 min, returned to 10% B at 5.5 min. The total running time was 10 min. Samples were analyzed in triplicates and three biological repeats were performed for each strain. Quantitation was based on the *m/z*, 565.0489 for UDP-glucose, 535.0372 for UDP-xylose, 579.0276 for UDP-glucuronic acid, and 188.0116 for $^{13}C_6$-Saccharin and expressed as relative levels to the internal standard $^{13}C_6$-Saccharin.

**Microscopy**. Zeiss observer fluorescent microscope equipped with an AxioCam digital camera was used to take differential interference contrast microscopy (DIC) and fluorescent images. The exposure times and other microscope settings were held constant between compared images. We employed the Surface function in the Imaris software (Bitplane Scientific Software, USA) to quantify the fluorescence intensity of cells expressing Fcy2-mNG. The fluorescence intensity was measured from three different images of each strains and at least 275 cells of each strain were evaluated. The intensity mean of each cell was imported into Graph Pad Prism (version 8.4.3) and analyzed.

**5-FC accumulation assay**. The cells were grown in YPD to log phase, washed and then transferred to YNB media and incubated for additional 3 h at 30 °C. Cells were harvested, washed with cold YNB without glucose and adjusted to 10 OD/ml. Each 100 µl of cells were mixed with 100 µl of 2× YNB media with glucose supplemented with 200 µM flucytosine and 0.8 µCi of [$^3$H] flucytosine (Moravek Inc; 4.0 Ci/mmole, 1 mCi/ml). Cells were incubated at 30° C in a water-bath shaker and 50 µl aliquots of sample were removed at 0, 1, 2, 5, 10, 20, 30, and 60 min and added to 2 mL cold media containing 20 mM flucytosine. Cells were immediately filtered with

24 mm Whatman GF/C filter and washed with cold YNB-A-N and assayed for radioactivity.

**Protein extraction and immunoblot analysis**. Total proteins were extracted with NaOH method[27]. An equal amount of protein (20 µg) was loaded and run on the Any kD Criterion TGX Stain-Free gel (Bio-Rad, Richmond, CA). The western blot was incubated with anti-FLAG antibody and developed using the Clarity Western ECL (Bio-Rad, Richmond, CA). The signal was quantitated using ChemiDoc MP imaging system (Bio-Rad, Richmond, CA) and band intensities were normalized to stain-free blots to control for loading.

**Statistical analysis**. Statistical analyses were performed using Graph Pad Prism (version 8.4.3). Student's *t* test and ANOVA followed by multiple-comparison test were used to determine whether there were significant differences between indicated samples.

**Reporting summary**. Further information on research design is available in the Nature Research Reporting Summary linked to this article.

## Data availability
The sequencing data from this study have been submitted to the NCBI BioProject under the accession PRJNA704667. FungiDB is available at [https://fungidb.org/fungidb/app]. All relevant data are available upon request. Source data are provided with this paper.

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

## Acknowledgements

We thank Kazuhiro Ohshima and Nathan Kwon for technical assistance. The authors thank Brigit Shea Sullivan of NIH Library Editing Service, for manuscript editing assistance. Deletion strains of KN99α and H99 were obtained from the Fungal Genetics Stock Center (Manhattan, Kansas, USA). This work was supported by the Division of Intramural Research (DIR), NIAID, NIH.

## Author contributions

J.K.-C. and J.E.B. initiated the project. Y.C.C. designed the experiments. Y.C.C. and A.K.L. performed the experiments. H.C. and P.J.W. performed the nucleotide sugar measurements. Y.C.C. and J.K-C wrote the manuscript. All authors reviewed the manuscript.

## Funding

## Competing interests

The authors declare no competing interests.
