## [Peer Review File · Nature Communications]

REVIEWER COMMENTS

Reviewer #1 (Remarks to the Author):

This manuscript reports studies of mechanisms of resistance to 5-fluorocytosine in the fungal pathogen *Cryptococcus*. The experiments are thorough and careful (including testing multiple strains, examining suppressors, and recapitulating multiple mutations of interest), methods are clear, the results are well-presented, and most studies support the conclusions drawn (minor exceptions noted below). Some of the data presented nicely supports or extends prior observations in the field, including confirmation that genes observed to mediate 5-FC toxicity in other yeasts behave similarly in the strains tested; suggestion that aneuploidy may influence drug resistance (modestly in this context); and demonstration of the levels of UDP-GlcUA in the presence of various mutations.

New findings in this work include the high frequency of resistance, a potential modest contribution of *URA6* to 5-FC resistance, and the demonstration that in cells with increased levels of UDP-GlcUA the permease *Fcy2* increases (supported both by direct studies and drug uptake measurements). The first two of these areas are documented but not pursued in detail, although feasible experiments to address the high frequency of resistance are suggested in the Discussion. The last is an interesting finding, since although it has previously been shown that increased levels of UDP-GlcUA yield resistance to 5-FC, it was not clear how this occurred. Unfortunately, there is no discussion or exploration of how UDP-GlcUA might impact permease expression.

Specific comments

1. The title suggests the paper is about the mechanisms by which high-frequency resistance develops, but that is not addressed.
2. 163-6 is overstated based on data in refs 19 (*FUR1*) and 29 (*FCY2*).
3. The strain table should be referenced early in the text as it is required to understand the figures.
4. 280-3 Given the known roles of *Nrg1* and *Ugd1*, why did the authors consider that these suppressors would improve growth of cells lacking *Uxs1*?
5. Fig. 7 would benefit from clearer microscopy (if not attempts to identify the subcellular structures noted) and some form of intensity quantitation – one image is not enough to support the statement of lower fluorescence.
6. Figures S3 and S4 should be presented in the results, rather than new data being introduced in the Discussion.
7. How do the authors explain the *AFR1* results (modest increase in resistance with an extra copy but no change upon deletion)? Could the marker in the engineered strains make a difference here?
8. 391 It is clear that the *UGD-G19A* mutant is functional in terms of producing UDP-GlcUA, but the UDP-GlcUA levels should not be interpreted as reflecting synthesis alone, since when *Uxs1* is present this compound is efficiently consumed. The mutant protein is likely to be highly inefficient in activity, based on the results in the double mutant, where it is only slightly above wild type.
9. The *UGD-G19A* mutant cells appear to have reduced capsules in both backgrounds compared to wild type; quantitation would be helpful here.
10. 413 states that it isn't clear how the suppressors change the *uxs1* phenotype, but it seems quite clear from prior literature on *Nrg1* targets and *Ugd1* biochemical function – am I missing something here?
11. 484 The method that worked poorly and therefore wasn't used can be omitted
12. Fig 6 legend title: High and low are switched

Reviewer #2 (Remarks to the Author):

The authors demonstrate that *Cryptococcus* exhibit resistance to 5 fluorocytosine at a high frequency when exposed to several fold above the MIC concentration of isolates. They identified that Chromosome 1 aneuploidy or duplication of ABC transporter gene only slightly increased the

resistance to 5FC, however the mutations in FCY2, FCY1, FUR1, UXS1 and URA6 contributed to the resistance in 5FC. Unlike well-studied multiple resistance mechanisms to fluconazole in *Cryptococcus* and *Candida*, 5 fluorocytosine resistance in *Cryptococcus* has not been defined well. The study provides insight in the development of 5 FC resistance in invitro settings. Concerning 5FC is a significant drug in induction therapy of cryptococcosis the data presented opens avenues for exploration of mechanisms concerning resistance to 5 FC in clinical settings.

There are certain issues that warrant clarifications

The different strains of *cryptococcus* analysed showed variations in the mechanism of development of 5FC resistance for example increase in the copy number of transporter gene AFR1 resulted in increase in the MIC of 5FC in R265 strain but not in WM 276. Similarly increased chromosome copy number was not found among the sequenced drug resistant clones derived from R265. Overall, *cryptococcus* displayed mechanisms independent of mutations in FCY2, FCY1, FUR1, UXS1 and URA6 that can decrease susceptibility to 5FC. It remains speculative that these "alternative" mechanisms may confer clinical resistance. How do the authors convince that mutation to resistance was preceded by mutation in a different gene that conferred relatively small increase in MIC.

Further, aneuploidy could be demonstrated to cause partial increase in MIC in some strains suggesting that chromosome 1 aneuploidy has a limited role in development of resistance. It would be interesting to know if the susceptibility to other antifungals were altered in Ch1 aneuploidy strains.

The discussion section can be improved by highlighting how the findings are extrapolated in clinical settings. Is it possible that exposure to 5 FC can induce such changes in vivo especially while on long term 5FC therapy?

Lines 120-121. Please clarify why fluconazole heteroresistant strains were included to determine 5 FC resistance.

Finally, authors may consider summing up the findings in the allied approach. As understanding the regulation of 5FC drug susceptibility in *cryptococcus* by chromosome aneuploidy, the regulation of FCY2, FCY1, FUR1, UXS1 and URA6 genes, additional factors involved in the complex regulation of genes on aneuploid chromosomes, and still unidentified genes remains to explored.

Reviewer #3 (Remarks to the Author):

This study confirms, by using WGS and mutant construction, the role of FCY1, FCY2 and FUR1 genes in the resistance of 5CF for *Cryptococcus neoformans* and *gattii*. The authors also show the putative role of URA6 gene in the 5FC resistance, and they also found that UXS1 gene is involved in the accumulation of 5FC into the cryptococcal cell, and so correlated with the susceptibility of 5FC, via modulation of the Fcy2 expression levels. This study is interesting for the understanding of the mechanism of antifungal resistance, this is well written and clear. Some minor modifications and details should be added.

Minor revisions

Line 52 : Could you please clarify the description of the 12 isolates because you said that there is 23 patients and 10 respond to flucytosine monotherapy, so 1 patient is missing.

There is confusing informations because Line 26 in the abstract you indicate that « *Cryptococcus* exhibits resistance to 5-FC at a high frequency when exposed to concentrations several fold above the minimal inhibitory concentrations of isolates" meaning high concentrations but in the Line 80 "5-FC resistant colonies emerge (...) at moderate concentrations of 5-FC." What is a moderate concentration?

In the MM section you indicate that drug concentration ranging 32 to 0.06 => but when we read the results section and the figure, apparently MIC 5FC for H99 should be 0.025 (because the contraction 20 fold above the MIC is 0.5 in figure 1). In figure 1 and S1 resistance seems to emerge at low concentration, could you please clarify the MIC value.

Figure 2, colors are too light and difficult to see correctly. It is not clear on which chromosome, chr

1 or 12?

Line 129 MIC 5FC for H99 is 2µg/ML but it doesn't correspond to MIC in the previous paragraph. Was the Fluconazole MIC determined using the same method ? indicate in MM

Table S1 and S2 is not indicated in the excel file, legend are missing for Tables S. Table S4 has a different police

Line 490 more technical details could be useful, shaking, did you test different 96 plates with different bottom form for example? What is the OD value for growth rate measurement?

Line 33 « in both in « remove the second « in »

Line 337 if experiments have been performed with *S. cerevisiae* they should be explained in the MM and results section and information is not relevant in the discussion part.

Line 347 It is known that 5FC induce mutation and failure of treatment that's why monotherapy is not recommended, this information is in the introduction and should may be also remind here.

Line 394 KN99 instead of Kn99

Reviewer #1 (Remarks to the Author):

.... New findings in this work include the high frequency of resistance, a potential modest contribution of URA6 to 5-FC resistance, and the demonstration that in cells with increased levels of UDP-GlcUA the permease Fcy2 increases (supported both by direct studies and drug uptake measurements). The first two of these areas are documented but not pursued in detail, although feasible experiments to address the high frequency of resistance are suggested in the Discussion. The last is an interesting finding, since although it has previously been shown that increased levels of UDP-GlcUA yield resistance to 5-FC, it was not clear how this occurred. Unfortunately, there is no discussion or exploration of how UDP-GlcUA might impact permease expression.

Response: Since it is possible that high levels of UDP-GlcUA might affect the expression of permease by altering the expression of transcription factor(s), we have used RNAseq analysis to compare the transcriptome profiles among the wild-type strain, the *uxs1Δ* mutant, and suppressor containing strains in the *uxs1Δ* and wild-type background. However, we did not find a probable transcription factor gene that might affect the permease expression. It is possible that accumulation of UDP-GlcUA might affect the Fcy2 levels at the translational or post-translational steps. We did not pursue the possibility, because to dissect the mechanism of how UDP-GlcUA accumulation regulates Fcy2 in detail is a separate project in itself. Not only will it take long time but we consider it to be beyond the scope of the current study.

Although we do not know how UDP-GlcUA might impact permease expression, we demonstrated clearly that the mechanism of 5-FC resistance in the *uxs1Δ* mutants is through the accumulation of UDP-GlcUA which decreases the permease levels that reduce 5-FC uptake. The *uxs1Δ* suppressor strains have lower Ugd1 function or *UGD1* expression which encodes the enzyme converting UDP-glucose to UDP-GlcUA, which in turn reduces the accumulation of UDP-GlcUA.

We have added the following sentences in the discussion (Lines 599-606) as the reviewer suggested. "It is not clear how exactly UDP-GlcUA accumulation might impact permease expression. It is possible that high levels of UDP-GlcUA might affect the expression of permease by altering the expression of transcription factor(s). However, we have failed to identify a probable transcription factor gene through analysis of transcriptome profile using the RNAs isolated from the wildtype, the WM276 *uxs1Δ* mutant, and suppressor containing strains in the *uxs1Δ* and wild-type background (data not shown). It is also possible that accumulation of UDP-GlcUA might affect the Fcy2 levels at the translational or post-translational steps. The detailed mechanism(s) of how UDP-GlcUA might affect permease expression remains to be elucidated."

Specific comments

1. The title suggests the paper is about the mechanisms by which high-frequency resistance

develops, but that is not addressed.

Response:

We have changed the title to “Moderate levels of 5-fluorocytosine cause the emergence of high frequency resistance in cryptococci”

2. 163-6 is overstated based on data in refs 19 (*FUR1*) and 29 (*FCY2*).

Response:

We have modified the sentences as “..., their roles have not been unequivocally substantiated for cryptococcal 5-FC resistance²⁸ except that *FCY2* in R265 and *FUR1* in H99 have been shown to be important for 5-FC resistance^{19,29}. (Line 206)

3. The strain table should be referenced early in the text as it is required to understand the figures.

Response:

We have added “Supplementary Table 1” early in the result section in Line 140.

4. 280-3 Given the known roles of *Nrg1* and *Ugd1*, why did the authors consider that these suppressors would improve growth of cells lacking *Uxs1*?

Response:

The suppressors of *uxs1Δ* mutants, *nrg1* and *ugd1*, were identified by repeated transfers of the *uxs1Δ* mutants in nonselective medium. Therefore, it is possible that the growth rate has been improved in the suppressor containing clones during the repeated transfer. We showed that the growth rate of the *uxs1* mutants containing *UGD1^{G19A}* suppressor was improved in the WM276 background, but it was not the case with the KN99α *uxs1Δ* mutant containing either *nrg1* or *ugd1* suppressors. It is not clear how the *UGD1^{G19A}* suppressor influence the growth in WM276 background. Since the presence of the suppressors reduced the accumulation of UDP-GlcUA, it is possible that such a reduction could also restore the growth defect of *uxs1Δ* mutant in WM276. However, it is unclear how repeated transfers of the KN99α *uxs1Δ* strains in drug-free media were selective for the suppressor containing cells with no advantage on growth rate. One possible scenario could be that deletion of *UXS1* in KN99α may reduce the life span of cells and the suppressor containing cells could overcome the deficiency. Therefore, the *uxs1* mutant cells are outnumbered by the suppressor containing cells during repeated transfers. We modified the discussion in Lines 505-597.

5. Fig. 7 would benefit from clearer microscopy (if not attempts to identify the subcellular structures noted) and some form of intensity quantitation – one image is not enough to support the statement of lower fluorescence.

Response:

As suggested by the reviewer, we quantified the fluorescence intensity in the images. We have added the followings in the material and methods section. “We employed the Surface function in the Imaris software (Bitplane Scientific Software, USA) to quantify the fluorescence intensity of cells expressing Fcy2-mNG. The fluorescence intensity was measured from three different images of each strain which included at least 275 cells. The intensity mean of each cell was imported into Prism 7 and analyzed.”. The data has been presented in Fig. 8b.

6. Figures S3 and S4 should be presented in the results, rather than new data being introduced in the Discussion.

Response:

As suggested by the reviewer, we have moved the description and Fig. S4 to the result section (Line 359-386) and changed the figure name as Fig. 6.

Since the information about *S. cerevisiae* is irrelevant to the discussion, we have deleted both the information and data presented in Fig. S3.

7. How do the authors explain the *AFR1* results (modest increase in resistance with an extra copy but no change upon deletion)? Could the marker in the engineered strains make a difference here?

Response:

We used the same neomycin resistant gene cassette containing *ACTIN* promoter and *TrpC* terminator as a selectable marker to construct the *afr1* deletion strains as well as the strains containing 2 copies of *AFR1*. The extra copy of *AFR1* was inserted in the region of chromosome 3 in H99 which does not contain any known transcript. Therefore, the marker in the engineered strains is unlikely to make a difference. *Afr1* is an ABC transporter which is known as an azole efflux pump. However, *Afr1* may have a minor role in transporting other xenobiotics such as 5-FC and extra copy of *AFR1* contributes modestly to the 5-FC resistance in H99.

We have modified the discussion in Lines 487-496 as the followings. “Interestingly, an extra copy of the ABC transporter gene, *AFR1*, slightly increases the drug resistance levels in H99. Although *Afr1* is known as a primary azole efflux pump and deletion of *AFR1* has no clear impact on the level of 5-FC resistance⁴⁹, *Afr1* might have a supplementary minor role in transporting other xenobiotics such as 5-FC in combination with other transporters. Thus, extra copy of *AFR1* could contribute modestly to the 5-FC resistance in H99.”

8. 391 It is clear that the UGD-G19A mutant is functional in terms of producing UDP-GlcUA, but the UDP-GlcUA levels should not be interpreted as reflecting synthesis alone, since when *Uxs1* is present this compound is efficiently consumed. The mutant protein is likely to be highly inefficient in activity, based on the results in the double mutant, where it is only slightly above wild type.

Response:

We agree with the reviewer and have modified the section as the following:

“Unlike the loss-of-function mutations in Ade13 suppressors, the *UGD^{G19A}* variant is functional in terms of producing UDP-GlcUA (Fig. 5). However, the activity of mutant protein may have been suffered since the amounts of UDP-GlcUA in the suppressor strain of the *uxs1Δ* mutants is not much higher than that of the wild type. In addition, while the *ugd1Δ* mutant was acapsular, the *UGD^{G19A}* containing strains were encapsulated in both KN99α and WM276 backgrounds (Fig. 6). But the capsule size in the *UGD^{G19A}* containing strains was slightly smaller than that of the wild type in both KN99 and WM276 suggesting that *UGD^{G19A}* variant may have lost some of its functions. The second type of *uxs1* suppressor, *NRG1*, was found partially deleted in KN99α. *NRG1* encodes a transcription factor which has been shown to regulate the expression of *UGD1*⁴⁸. It is possible that the action of suppression by *Knrg1* is through downregulation of *UGD1* since we found the expression levels of *UGD1* were reduced in *Knrg1* containing strains compared to the wild type (Fig. 6d). Therefore, the Ugd1 function or the *UGD1* expression was reduced in the suppressor containing strains which may result in the decrease of the UDP-GlcUA accumulation.” As suggested by the review in point 6, we have moved these descriptions to the result section in Lines 369-386.

9. The UGD-G19A mutant cells appear to have reduced capsules in both backgrounds compared to wild type; quantitation would be helpful here.

Response:

We agree with the reviewer and the quantitation of capsule size is provided. We have added the followings in the Fig. 6 legend. “The capsule size of more than 65 cells were measured from 5 different fields for each culture. Capsule thickness was defined as the distance between the visualized cell wall and the edge of the capsule. Results were presented as mean ± SD (**** = $p < 0.0001$).” As suggested by the reviewer in point 6, we have moved the data to the Lines 376-380 in result section.

10. 413 states that it isn’t clear how the suppressors change the *uxs1* phenotype, but it seems quite clear from prior literature on Nrg1 targets and Ugd1 biochemical function – am I missing something here?

Response:

We have modified the statement as the following: “It is not clear how exactly UDP-GlcUA accumulation might impact permease expression.” (new Line 599). We continued the discussion after this sentence as described above in the response to the general concern of this reviewer.

11. 484 The method that worked poorly and therefore wasn’t used can be omitted

Response:

We have removed the section as suggested by the reviewer.

12. Fig 6 legend title: High and low are switched

Response:

The typos have been corrected.

Reviewer #2 (Remarks to the Author):

There are certain issues that warrant clarifications

1. The different strains of cryptococcus analyzed showed variations in the mechanism of development of 5FC resistance for example increase in the copy number of transporter gene AFR1 resulted in increase in the MIC of 5FC in R265 strain but not in WM 276. Similarly increased chromosome copy number was not found among the sequenced drug resistant clones derived from R265. Overall, cryptococcus displayed mechanisms independent of mutations in FCY2, FCY1, FUR1, UXS1 and URA6 that can decrease susceptibility to 5FC. It remains speculative that these “alternative” mechanisms may confer clinical resistance. How do the authors convince that mutation to resistance was preceded by mutation in a different gene that conferred relatively small increase in MIC.

Response:

We showed that the laboratory created chr1 aneuploidy only modestly increased the 5-FC resistance while the original HL8 and HS2 isolates containing chr1 disomy was highly resistant. It is possible that other unverified changes in the genome of HL8 and HS2 contributed to the 5-FC resistance. However, we do not know the order of appearance of different mutations in these isolates. We can only speculate that at moderate concentrations, 5-FC may not sufficiently inhibit cell growth and the metabolized 5-FC can cause various modifications including chromosome aneuploidy. Subsequently cells containing modifications in genes relevant for 5-FC resistance are selected and form resistant colonies at high frequency. We have discussed this possibility on Lines 438-442.

2. Further, aneuploidy could be demonstrated to cause partial increase in MIC in some strains suggesting that chromosome 1 aneuploidy has a limited role in development of resistance. It would be interesting to know if the susceptibility to other antifungals were altered in Ch1 aneuploidy strains.

Response:

We have tested the chr1 aneuploid strains for their susceptibility to Terbinafine, Fenpropimorph and Gliotoxin but found no increase in resistance. Since they were not relevant for the scope of present study, the data have not been included.

3. The discussion section can be improved by highlighting how the findings are extrapolated in clinical settings. Is it possible that exposure to 5 FC can induce such changes in vivo especially while on long term 5FC therapy?

Response:

We have added the followings in the discussion (Lines 471-476). “It is difficult to extrapolate our *in vitro* results in the clinical settings since the 5-FC dosage recommended for therapy is based on the body weight of patients while our study was conducted on agar plates at 30°C. Although it has been recommended that the 5-FC dose in patients be adjusted to maintain the serum 5-FC levels at 50 to 100 µg/ml ⁴⁶, possibility still exists that 5-FC can induce such changes *in vivo* during long term 5FC therapy and cause treatment failure.”

4. Lines 120-121. Please clarify why fluconazole heteroresistant strains were included to determine 5 FC resistance.

Response:

We have modified the sentences as followings. “Since the copy number of chr1 was elevated in HL8 and HS2, chr1 aneuploidy was presumed to have contributed to 5-FC resistance. This assumption was based on the observation that heteroresistant to 32 µg/ml fluconazole contained duplicated chr1 ²⁵. We generated two H99 derived heteroresistant clones, C1976 and C1977 (Table S1), which were supposed to be disomic for chr1 and used them to determine the effect of chr1 duplication for 5-FC resistance.” (in Lines 134-141).

5. Finally, authors may consider summing up the findings in the allied approach. As understanding the regulation of 5FC drug susceptibility in cryptococcus by chromosome aneuploidy, the regulation of FCY2, FCY1, FUR1, UXS1 and URA6 genes, additional factors involved in the complex regulation of genes on aneuploid chromosomes, and still unidentified genes remains to explored.

Response:

We have added the following sentences in Lines 478-482.

“In understanding the regulation of 5-FC drug susceptibility in cryptococci, we identified several factors that contribute to the change of susceptibility including mutations in *FCY2*, *FCY1*, *FUR1*, *UXS1* and *URA6* genes in addition to several unverified variants that remains to be explored; developing of suppressors; chromosome aneuploidy and the regulation of the gene on aneuploid chromosome.”

Reviewer #3 (Remarks to the Author):

Minor revisions

1. Line 52: Could you please clarify the description of the 12 isolates because you said that there is 23 patients and 10 respond to flucytosine monotherapy, so 1 patient is missing.

Response:

We have modified the sentence in Line 56-58 as “From the 13 failing patients, isolates were available for study from 12 patients, among which six isolates were highly resistant to 5-FC.”

2. There is confusing information because Line 26 in the abstract you indicate that « Cryptococcus exhibits resistance to 5-FC at a high frequency when exposed to concentrations several fold above the minimal inhibitory concentrations of isolates” meaning high concentrations but in the Line 80 “5-FC resistant colonies emerge (...) at moderate concentrations of 5-FC.” What is a moderate concentration?

Response:

We have modified the sentence in Line 89 as “5-FC resistant colonies emerge at high frequency at concentrations several folds above the MIC”

3. In the MM section you indicate that drug concentration ranging 32 to 0.06 => but when we read the results section and the figure, apparently MIC 5FC for H99 should be 0.025 (because the concentration 20 fold above the MIC is 0.5 in figure 1). In figure 1 and S1 resistance seems to emerge at low concentration, could you please clarify the MIC value.

Response:

The 32µg/ml to 0.0625µg/ml is the range of drug concentrations used for determination of MIC by broth method. As described in the Materials and Methods, we used two methods to determine the MIC of the strains. Unless specified, the 5-FC MICs were determined by using the Liofilchem® MTS™ (MIC Test Strips) on the agar plates. When MIC levels of mutant strains were close to the wild type, we used the broth MIC method recommended by CLSI M27-A3 to verify the MIC levels and the range was 32µg/ml to 0.0625µg/ml. The different outcomes of the MIC from the two methods are because one was performed on agar YNB medium at 30°C and the other was in the RPMI 1640 liquid medium at 37°C. The H99 MIC determined by MIC Test Strips was between 0.023 - 0.032 µg/ml as shown in Table 1 and 2 which is similar to the calculated value 0.025 µg/ml described by the reviewer.

4. Figure 2, colors are too light and difficult to see correctly. It is not clear on which chromosome, chr 1 or 12?

Response:

We have modified the figure legend to clarify the point as the following: “The change of copy number is represented by the intensity of the color. The red color represents copy gain and blue color represents copy loss in units of log₂ as shown by the color scale”. Since the color intensity reflects the copy number, the color is either light or invisible when the chromosome copy number is close to the reference. There were only a few clones which had clear changes of chromosome copy number. In addition, we have modified the labeling of chromosome number in the bottom H99 panel.

5. Line 129 MIC 5FC for H99 is 2µg/ML but it doesn't correspond to MIC in the previous paragraph. Was the Fluconazole MIC determined using the same method ? indicate in MM

Response:

The 2µg/ml 5-FC MIC was determined by broth microtiter test as indicated in a new Line 133-134. The 5-FC MIC in previous paragraph was determined by the Liofilchem MIC Test Strips. We have clarified the method for fluconazole MIC determination in Materials and Method as the followings (Lines 639-641). “Briefly, 5-FC or fluconazole was dispensed in 96-well microtiter plates with 2-fold serial dilutions of drug concentrations ranging from 32µg/ml to 0.0625µg/ml in MOPS-buffered RPMI 1640 and incubated at 37°C for 72 hrs.”

6. Table S1 and S2 is not indicated in the excel file, legends are missing for Tables S. Table S4 has a different police

Response:

For convenience, we made a new file “**Description of Additional Supplementary Files**” to describe the content of the files and changed the name of Table S1 and S2 to Supplementary Data 1 and 2.

7. Line 490 more technical details could be useful, shaking, did you test different 96 plates with different bottom form for example? What is the OD value for growth rate measurement?

Response:

The input OD was 0.05 and growth measurement was performed with setting at double orbital, 2mm fast with 548 cpm. The reading was taken at 15min intervals for 24 h as described in Lines 697-716. We have added “The Max V was obtained from log phase using Gen5 Data Analysis Software (Biotek, Winooski, VT) and the doubling time was calculated by $\text{LN}(2)/\text{Max V}$.” We tested both round and flat bottom 96 well plates and both types of plates failed to yield satisfactory results. As suggested by reviewer #1, we have deleted the description of 96 wells experiments.

8. Line 33 « in both in « remove the second « in »

Response:

We have removed the second “in”.

9. Line 337 if experiments have been performed with *S. cerevisiae* they should be explained in the MM and results section and information is not relevant in the discussion part.

Response:

We agree with the reviewer that the information on *S. cerevisiae* is irrelevant. We deleted the information on *S. cerevisiae* and Figure S3.

10. Line 347 It is known that 5FC induce mutation and failure of treatment that’s why monotherapy is not recommended, this information is in the introduction and may be also remind here.

Response:

Instead of reverberating the statement, we have added the followings in the discussion (Lines 471-476). “It is difficult to extrapolate our in vitro results in the clinical settings since the 5-FC dosage recommended for therapy is based on the body weight of patients while our study was conducted on agar plates at 30°C. Although it has been recommended that the 5-FC dose in patients be adjusted to maintain the serum 5-FC levels at 50 to 100 µg/ml ⁴⁶, possibility still exists that 5-FC can induce such genomic changes in vivo while patients are on long term 5FC therapy resulting in treatment failure.”

11. Line 394 KN99 instead of Kn99

Response:

We have corrected the typo.

REVIEWERS' COMMENTS

Reviewer #1 (Remarks to the Author):

The authors have thoughtfully addressed the review comments.

Reviewer #3 (Remarks to the Author):

This study confirms, by using WGS and mutant construction, the role of FCY1, FCY2 and FUR1 genes in the resistance of 5FC for *Cryptococcus neoformans* and *gattii*. The authors also show the putative role of URA6 gene in the 5FC resistance, and they also found that UXS1 gene is involved in the accumulation of 5FC into the cryptococcal cell, and so correlated with the susceptibility of 5FC, via modulation of the Fcy2 expression levels. This study is interesting for the understanding of the mechanism of antifungal resistance, this is well written and clear. Authors answered to different remarks of reviewers and added some interesting informations in the discussion section.